# GIS-Based Modeling of Human Settlement Suitability for the Belt and Road Regions

**DOI:** 10.3390/ijerph19106044

**Published:** 2022-05-16

**Authors:** Wenjun Li, Peng Li, Zhiming Feng, Chiwei Xiao

**Affiliations:** 1Institute of Geographic Sciences and Natural Resources Research, Chinese Academy of Sciences, Beijing 100101, China; liwj.17s@igsnrr.ac.cn (W.L.); lip@igsnrr.ac.cn (P.L.); xiaocw@igsnrr.ac.cn (C.X.); 2College of Resources and Environment, University of Chinese Academy of Sciences, Beijing 100049, China; 3Key Laboratory of Carrying Capacity Assessment for Resource and Environment, MNR, Beijing 101149, China

**Keywords:** Geographic Information System, index model, human settlement suitability, the Belt and Road

## Abstract

(1) Background: Human settlements are a basic requirement for human survival and development. The natural suitability of human settlements directly affects human health and their quality of life and, to some extent, also dictates the size of the regional population and economy, as well as the level of urbanization. (2) Methods: This study presents a kilometer grid-based comprehensive human settlement suitability (HSS) evaluation model (containing the relief degree of the land surface (RDLS), the temperature–humidity index (THI), the land surface water abundance index (LSWAI), and the land cover index (LCI)). (3) Results: Based on the correlations between the four factors (i.e., the normalized RDLS (NRDLS), normalized THI (NTHI), LSWAI, and LCI), the NRDLS and NTHI were determined to be the leading factors, and the LSWAI and LCI were considered to be the auxiliary factors. The auxiliary factors were used to enhance the comprehensive HSS model determined by the leading factors. Based on this logic, spatial and index models were established separately. The HSS index for the BRI regions ranged from 0.07 to 1.00. Six levels of HSS were determined—namely, unsuitable, critically unsuitable, critically suitable, generally suitable, moderately suitable, and highly suitable conditions. In particular, the human settlements suitability is dominated by critically suitable and suitable conditions that accounted for 63% of the BRI. (4) Conclusions: The evaluation results of human settlements showed that topographic and climatic conditions are important limiting factors for the suitability of human settlements. Due to the renewability and manmade adjustability of hydrological and land cover conditions, the comprehensive suitability of human settlements shows differences in different geographic spaces along the BRI. The results provide a decision-making basis for the research on the resource carrying capacity and reasonable distribution of populations in the BRI, so as to realize the sustainable development along the regions.

## 1. Introduction

The natural environment of human settlements exerts a significant impact on the development of human society [1] and, to a certain extent, dictates the spatial population distribution pattern [2]. Evaluating the natural suitability of human settlements can help to not only scientifically determine human settlement conditions, which are affected by natural factors, but can also characterize regional population distribution patterns, which are affected by natural limitations or settlement suitability. Simultaneously, the results provide decision-making support for predicting population sizes, determining functional zones of population development, and ensuring sustainable development [3]. In recent decades, human settlement suitability (HSS) evaluation models have been extensively applied in China in the national planning of the main functional zones [4] and have been used in studies of regional population-carrying capacities [5]. Furthermore, they are also applied in the population, resource, environment, and development fields [6]. The existing HSS methods use a simple model of single-element grading and the spatial superposition of single elements for comprehensive grading. The previous study methods did not consider the influence degree and interaction relationship of the elements in the evaluation model. The world, particularly the regions involved in the Belt and Road Initiative (BRI), has been undergoing sustained and steady economic development. The increasingly high levels of human activity have had a profound impact on the regional climate and natural environments [7,8]. Among them, land cover, as well as vegetation and hydrological conditions, are particularly prominent [9]. Understanding human settlements in the countries and regions along the BRI under the new normal conditions amid the changes in the natural environmental factors is of great significance, as it can provide the basis for making decisions on BRI construction and a reasonable population distribution [10]. 

There is a large geographical area along the BRI routes. The countries along the BRI routes have different natural resources and are relatively highly economically developed [11]. There is a great potential for cooperation between these countries. Additionally, the regions involved in the BRI are situated in a zone sensitive to climate change, with complex natural conditions and diverse and vulnerable natural environments [12]. The countries in these regions generally face a conflict between development and environmental protection. Investigating human settlement conditions in the countries along the BRI and determining their current water and land resources and population distribution patterns are of great practical significance to the realization of sustainable development [13]. This study is based on the 17 goals of the United Nations 2030 Agenda for Sustainable Development, with particular emphasis on meeting everyone’s life needs in terms of health, safety, and sanitation. When natural conditions cannot satisfy everyone in terms of climate environment, water use, food security, etc., it is necessary to strengthen cooperation and promote technological innovation to improve the limitations of the settlement environment caused by the natural geographical factors. Therefore, before this, it is necessary to clarify the degree of influence of different natural and geographical factors on the human settlements environment and to improve the settlements environment of the local residents in a targeted manner according to local conditions so as to achieve the goal of allowing everyone to live a healthy life [14].

The study development of human settlements has gradually transitioned from the perspective of architecture and urban planning to the perspective of geography and combined with 3S technology to establish a grid-scale evaluation index for the natural suitability of human settlements. Among them, the evaluation of the natural suitability of human settlements based on topography, climate, hydrology, and land cover conditions is the most widely used. There has been little study on human settlements on a large scale and even more rarely in the Belt and Road regions. When it comes to HSS, the pioneering and exploratory research was first conducted to study level framework evaluation indices and models [5]. Later, cross-sectional research was gradually performed at natural and administrative division levels [15]. Additionally, the evaluation factors and models have been continuously adjusted in regional evaluation processes [16], such as the use of the temperature–humidity index (THI) or the wind effect index in the evaluation of climate suitability, such as the addition of natural disaster factors for regions at high risk of geological disasters [5]. Then, these indices and factors were extensively evaluated [17]. On the one hand, a more objective basis is needed to determine the key critical thresholds for classifying the suitability (unsuitable, critically suitable, generally suitable, moderately suitable, and highly suitable) in the HSS evaluation study [18]. On the other hand, comprehensive HSS evaluation methods also require more comprehensive and continuous practical investigations [19]. In summary, it is necessary to strengthen the HSS evaluation research in the new era at both the scientific research and practical ecological protection levels [20].

This study intends to address three scientific questions: (1) Find out the natural background conditions of the Belt and Road regions, such as topography, climate, hydrology, and land cover. (2) According to the degree of influence of the physical and geographical factors in the BRI regions for the human settlement environment and the correlation between the factors, this study built a human settlement suitability index model. (3) Based on the GIS platform, a raster atlas was formed for a comprehensive evaluation of human settlements in the BRI regions, and grading and a zonal evaluation of the human settlements in the study area were conducted, providing a research basis for the rational distribution of the population along the Belt and Road.

## 2. Materials and Methods

### 2.1. Materials

#### 2.1.1. Global Digital Elevation Model (GDEM) Data

In this study, 30-m Advanced Spaceborne Thermal Emission and Reflection Radiometer (ASTER)-Global Digital Elevation Model (GDEM) data (Version 2) were used as the elevation data. ASTER GDEM files covering all of the BRI regions were downloaded from the Japan Aerospace Exploration Agency website (http://gdem.ersdac.jspacesystems.or.jp/, accessed on 15 April 2022) and the U.S. National Aeronautics and Space Administration (NASA) website (http://reverb.echo.nasa.gov/reverb/, accessed on 15 April 2022). Each downloaded ASTER GDEM file contained two files: dem.tiff (i.e., digital elevation model data) and num.tiff (i.e., quality assurance data), resampled to 1 km for a large-scale study. 

#### 2.1.2. Monthly Ground Climate Datasets

The 1980–2017 monthly ground climate datasets acquired at meteorological stations in the BRI countries were obtained from the National Meteorological Information Center of China and were used as meteorological data in this study. The monthly data acquired at each station mainly contained meteorological data (including the monthly average temperature, relative humidity, and precipitation), as well as the longitude, latitude, and altitude of the station. The multi-year monthly average meteorological data in the study area were calculated by Kriging interpolation based on the GIS platform, using 2363 weather stations, with a spatial resolution of 1 km.

#### 2.1.3. Land–Surface Water Index and Normalized Difference Vegetation Index

The land–surface water index (LSWI) and normalized difference vegetation index (NDVI) data were obtained from the NASA EarthData Platform. The data in the MOD13A3 dataset (a Level 3 product) used in this study were sinusoidal projections that had already been subjected to correction treatments (positioning, calibration, and edge distortion correction). Additionally, the data resolution ratio was 1 km monthly and mid-infrared surface reflectance, which was collected by the Moderate Resolution Imaging Spectroradiometer (MODIS) Terra sensor. 

#### 2.1.4. Global Land Cover Data

The global land cover data used in this study were extracted from the section (containing 10 types of Level 1 land cover classes) of the Global 30-m Land cover Dataset (2017) published on the Sharing Service Platform of the National Earth System Science Data Center of China in September 2018. Additionally, the data are projections in the GCS_WGS_1984 geographic coordinate system, resampled to 1 km for a large-scale study.

### 2.2. Methods

This study developed a comprehensive HSS evaluation model for the countries and regions along the BRI based on four indices: the RDLS, the THI, the LSWAI, and the LCI. The single-factor indices are described in this section. The comprehensive evaluation model is described in this paper.

#### 2.2.1. RDLS

The topographic element of human settlements are expressed by RDLS:RDLS = ALT/1000 + {[Max(H) − Min(H)]} × {[1 − P(A)/A]}/500(1)

In the model, ALT is the average altitude within a certain region with a certain grid cell as its center (m), Max(H) and Min(H) are the maximum and minimum altitudes within the region, respectively (m), P(A) is the area of flat land (i.e., land with a relative height difference ≤ 30 m) within the region (km^2^), and A is the total area of the region (km^2^) [4].

#### 2.2.2. THI

The climate element of human settlements are expressed by THI:THI = 1.8t0.55 × (1 − f)(1.8t − 26)(2)

The THI represents the humidity-corrected temperature [21]. In the above equation, t is the monthly average temperature (°C), and f is the monthly average relative air humidity (%).

#### 2.2.3. LSWAI

The hydrological element of human settlements are expressed by LSWAI:LSWAI = α × P + β × LSWI(3)

In the model, P is the normalized annual average precipitation, α is the weight of the annual average precipitation in the study area, and β is the weight of the LSWI for the study area. α and β are set to 0.8 and 0.2, respectively, in this study.

#### 2.2.4. LCI

The land cover element of human settlements are expressed by LCI:LCI = NDVI × LT_i_(4)

The LCI characterizes the land use and land cover conditions in the study area. In the above equation, NDVI is the normalized difference vegetation index for the grid cell, and LT_i_ is the weight of each land use type. For example, the area around the grassland and agricultural land has a higher weight, and the bare land and ice/snow area has a lower weight. 

### 2.3. Details of the Study Area

The BRI is unique in terms of its geographic spatial layout. Taking into consideration the geographic patterns of both oceans and land, the BRI has developed a new geopolitical block with several developing countries as the main group [22] and the ancient Belt and Road that traversed the Eurasian continent as the major geographic axis. The countries along the BRI routes are distributed in various geographic zones. The regions differ relatively significantly in terms of their politics, economics, and culture, as well as their climate and water and land resources. The average altitude of the geographical area included in the BRI is 694 m (Figure 1a). Regions at altitudes below 200 m account for 39.25% of the BRI, which are primarily distributed in the plains in the middle and lower reaches of large rivers, e.g., the Middle–Lower Yangtze Plain, the Gangetic Plain, and the Indian River Plain. Regions at altitudes of 200~500 m account for 22.97% of the BRI, which are primarily distributed on the Deccan Plateau and Siberian Plateau in Russia. Regions at altitudes of 500~1000 m account for 17.01% of the BRI and are mainly distributed in the transition zone between the Arabian Plateau, Mongolian Plateau, and Northeast China Plain. Regions at altitudes of 1000~2000 m account for 13.11% of the BRI, which are primarily distributed in the Hengduan Mountains at the junction of the Persian Plateau, Mongolian Plateau, Loess Plateau, and Tibetan Plateau. Regions at altitudes above 2000 m account for 7.66% and are primarily distributed at the southern foot of the Himalayas and on the Northwestern Persian Plateau.

There are complex and variable climate conditions in the BRI, with an extremely uneven temperature distribution pattern from South Asia to Central and Eastern Europe [23]. The annual average temperature in the BRI area ranges from −17 to 29 °C. Regions with annual average temperatures below 10 °C account for 60.71% of the BRI (Figure 1b), which are primarily distributed in high-latitude and high-altitude zones, including Mongolia, Russia, and most countries in Central and Eastern Europe, as well as Northeastern China, the Inner Mongolian Plateau, and the Tibetan Plateau. Regions with annual average temperatures between 10 and 20 °C account for 15.70% and are primarily distributed at low altitudes between 25° N and 45° N. Regions with annual average temperatures higher than 20 °C account for 23.59% and are primarily distributed in tropical and subtropical zones, including Southern China, Southeast Asia, South Asia, and West Asia.

The multi-year average precipitation in the study area is approximately 400 mm (Figure 1c), which is lower than the annual average precipitation in China (approximately 628 mm), as well as the Asian and global average precipitation (approximately 740 and 834 mm, respectively). Arid, semiarid, subhumid, and humid regions account for 30.66%, 33.89%, 23.38%, and 12.17%, respectively. Regions with multi-year average precipitation below 100 mm are primarily located on the Arabian Peninsula, Southeastern Egypt, the Southwestern Mongolian Plateau, the northern edge of the Tibetan Plateau, Northwestern India, and in some zones within Central Asian countries. All the regions with average precipitation above 2000 mm are located in warm and humid Southeast Asia. With regards to land cover conditions, the proportions of forestland, bare land, and grassland in the study area are 28.07%, 23.51%, and 21.93%, respectively (Figure 1d). Forestland is primarily distributed in Russia, Southern and Northeastern China, and some European countries. Bare land is distributed on the Arabian Peninsula, Persian Peninsula, Mongolia Plateau, and other regions. Grasslands are centrally distributed in Northern Kazakhstan and Northern Mongolia, as well as near the 400-mm rainfall line of China. Additionally, croplands are spatially distributed on the plains, which accounts for approximately 14.96% of the study area. 

## 3. Results

### 3.1. Single-Factor Index Analysis and Factor Correlations of Human Settlements

#### 3.1.1. Single-Factor Index Analysis of Human Settlements in the BRI

The statistical analysis of the RDLS for the BRI regions showed that (Figure 2a): the RDLS was mainly low in the BRI regions. Specifically, the RDLS ranged from 0 to 19.9, with an average value of 1.05, and varied relatively significantly between regions. In the study area, the RDLS gradually decreased from the Himalayas, Tianshan Mountains, and Pamir Mountains to the surrounding regions. The RDLS was high in the central area and low in the peripheral regions. Spatially, low RDLS values were contiguous. Specifically, low RDLS values were concentrated on the East European Plain, West Siberian Plain, Northeast China Plain, North China Plain, Indian River Plain, and the Southern Indochinese Peninsula.

As shown in the results calculated using the THI model, the THI varied relatively significantly between the BRI regions. Except for the Tibetan Plateau, the THI gradually decreased from the low-latitude regions in the south to the high-latitude regions in the north. Extremely cold regions, with the THI below 40, were mainly located in high-latitude and high-altitude zones, including Northern Kazakhstan, Northeastern China, the Tibetan Plateau, the Tianshan Mountains, and the Pamir Mountains in Central Asia. Regions with a THI between 45 and 75, at which the human body feels relatively comfortable, were primarily distributed in most of the zones between 20° N and 45° N, except for the Tibetan Plateau, Pamir Mountains, and the surrounding high-altitude zones. Regions with a THI higher than 75 had a hot and humid climate, which were primarily distributed along the southeastern coast of Bangladesh and India, the Southern Indochinese Peninsula, and in most of the Malay Archipelago.

The average LSWAI was approximately 0.33, and those values were clustered in space for the BRI regions. Especially, high-LSWAI regions were primarily distributed in Southeastern China, Bangladesh, and Western Russia. Low-LSWAI regions were distributed in a concentrated, contiguous manner in the interior of the Asian continent located south of Russia and west of Western China. As shown in the results calculated using the LCI model (Figure 2d), the multi-year average LCI was approximately 0.21. Moreover, High-LCI regions were mainly distributed in Southwestern China, as well as on the main plains of the South Asian subcontinent (i.e., the Gangetic Plain and Indian River Plain). Low-LCI regions were primarily concentrated in two types of zones, namely, alpine zones (including the Tibetan Plateau, west of China’s 400-mm rainfall line, the Mongolian Plateau, and Northern Russia), as well as hot and dry zones (including the Persian Plateau, the Arabian Peninsula, and Egypt).

Additionally, referring to the classification criteria for the basic categories of topography and climatic elements, combined with the spatial distribution of the population in the study area, the terrain and climatic suitability of the human settlement environment is classified into six levels: namely, unsuitable, critically suitable, generally suitable, moderately suitable, highly suitable, and uninhabited. Table 1 summarizes the classification of suitability. 

#### 3.1.2. Correlation Analysis of Single-Factor Suitability Indices in the BRI

There are various factors affecting the suitability of human settlements, but the most fundamental and determining factors of other natural environments, the main factors that play a leading role in the natural environment of human settlements, mainly include the degree of terrain fluctuation, hydrothermal and climatic conditions, and hydrological conditions, as well as the characteristics of land use and vegetation coverage that comprehensively reflect the natural conditions of the region. According to the degree of influence of naturally geographical elements on human settlement environments, four types of constituent factors have been selected. The index system has been used experimentally in China. The results demonstrated that the four basic factors (RDLS, THI, LSWAI, and LCI) affecting human settlements could effectively characterize their comprehensive suitability. In this study, the same four single factors were also used; however, whether they affected the HSS to the same extent requires further investigation. Therefore, it was necessary to examine the positive (negative) synergistic effects of single factors at the early stage of modeling. Table 2 summarizes the statistical correlation analysis of the grid data for the whole BRI area.

Based on the four single HSS factors that constitute the comprehensive suitability of human settlements, six groups of correlation analyses were conducted between every two factors in the BRI regions, respectively. The results showed that, an extremely weak correlation was found between the RDLS and THI, such as the LSWAI and LCI. Additionally, based on prior knowledge, the terrain and climatic suitability were found to be the main determinants of the HSS. Therefore, the HSS was established, with the RDLS and THI as the leading factors, which are corrected by the LSWAI and LCI as auxiliary factors.

### 3.2. Construction of Comprehensive Evaluation Models for HSS

#### 3.2.1. Construction of a Spatial Evaluation Model for HSS in the BRI

The previous correlation analyses determined an extremely weak correlation between the NRDLS and the NTHI, such as the LSWAI and LCI. Thus, the weakly correlated factors were “independent” but not “influential”. Therefore, it was necessary to place the factors at “adjacent” locations that had moderate or stronger correlations with one another when constructing the comprehensive HSS evaluation model to take advantage of the correlations between the auxiliary and leading factors that affect or enhance the HSS. Based on this logic, a spatial model and an index model were constructed in this study to evaluate the HSS.

The HSS was evaluated within a square, with a side length of 1 km. Based on the correlation coefficient matrix of the four single factors affecting the HSS, the NRDLS and NTHI, which were two extremely weakly correlated factors, were placed at the diagonal corners of the square. These two factors were used as leading factors for evaluating the HSS. The LSWAI and LCI were used as the auxiliary factors. Additionally, the sides of the square were used to construct interlaced two-dimensional (2D) quadrants. The NRDLS and NTHI were used as the origin of the coordinates, and the LSWAI and LCI were used as the endpoint of the coordinates (endpoint value = 1). In Figure 3, the HSS consists of two evaluation results: namely, the terrain and climatic suitability corrected by the LSWAI and LCI conditions. Therefore, in this model, the HSS is comprehensively characterized by the sum of the areas of the two right triangles, one with the NRDLS as the starting point and one with the NTHI as the starting point. The area uncorrected by the LSWAI and LCI represents the unsuitable area for human settlement (HSUS). Hence, the theoretical value of the HSS index ranges from 0 to 1. Figure 3 shows a schematic diagram of the spatial logic model described above.

As demonstrated in Figure 3, a is the NRDLS corrected by the LSWAI (Normalization of the minimum effect: X = (X_i_ − X_min_)/(X_max_ − X_min_)), d is the NRDLS corrected by the LCI, b is the NTHI corrected by the LSWAI (Normalization of the central effect: X = (X_i_ − X_m_)/(X_m_ − X_min_) X_i_ < X_m_, X = (X_max_ − X_i_)/(X_max_ − X_m_) X_i_ > X_m_), and c is the NTHI corrected by the LCI. The index (physical) model for a comprehensive evaluation of human settlements is explained as follows.

#### 3.2.2. Construction of a Comprehensive Evaluation Index Model for HSS in the BRI

The comprehensive evaluation index model for HSS

The HSS, as shown in Equations (5)–(9):HSS = 0.5 × (NRDLS + r_h_)(NRDLS + r_v_) + 0.5 × (NTHI + t_h_)(NTHI + t_v_)(5)
r_h_ = LSWAI × R_th_ − NRDLS × LSWAI × R_th_
(6)
 r_v_ = LCI × R_tv_ − NRDLS × LCI × R_tv_(7)
t_h_ = LSWAI × R_ch_ − NTHI × LSWAI × R_ch_
(8)
t_v_ = LCI × R_cv_ − NTHI × LCI × R_cv_(9)

In the model, r_h_ is the hydrologically corrected terrain index, r_v_ is the cover-corrected terrain index, t_h_ is the hydrologically corrected climate index, t_v_ is the cover-corrected climate index, R_th_ is the correlation coefficient between the RDLS and LSWAI for the whole study area, R_tv_ is the correlation coefficient between the RDLS and LCI, R_ch_ is the correlation coefficient between the THI and LSWAI, and R_cv_ is the correlation coefficient between the THI and LCI.

Correction of the single-factor indices

The normalized results of the central effect of the THI deviate from the actual body sensations (Figure 4a), especially in Southeast Asia, where the NTHI is low. By analyzing the corresponding relationship between the NTHI and the THI, it can be seen that the THI ranged from 0 to 80. When THI = 65 is selected as the central effect value (the normalized value is 1), since the data volume and data distribution structure are not completely symmetrical, the normalized data structure is discrete in the region of THI > 65 (Figure 4b).

Therefore, smoothing based on the same interval threshold, as well as the relevant equations, was required. For example, the THI values of 45–50 and 70–75 belong to the types of generally suitable climates. Based on the above, the data whose THI value is greater than 65 is processed in segments. Figure 4c shows the results.
X_thic_ = −0.000144 × X_thi_ + 1.93899   65 < X_thi_ < 70(10)
X_thic_ = −0.000578 × X_thi_ + 4.97293   70 < X_thi_ < 75(11)
X_thic_ = −0.000268 × X_thi_ + 2.64894   75 < X_thi_ < 80(12)

X_thic_ and X_thi_ are the THI after the secondary correction and the original THI values, respectively.

Terrains and climate were determined to be the leading factors of the HSS when first constructing the model. Therefore, what the r_h_, r_v_, t_h_, and t_v_ corrects is its “unsuitable” part for the leading factors of the HSS. If using the auxiliary factors (i.e., the LSWAI and LCI) to correct the “suitable” part of the leading factors, this could conceal the background suitability and leading nature of the terrain and climate. Additionally, using the auxiliary factors to enhance the “suitable” part of the leading factors would result in an exponential increase in the calculated value of the HSS and underscore the impact of the LSWAI and LCI. This would render it impossible to determine the factors that cause regions to have high HSS values. In summary, in this model, the correction results can control the calculation results of the comprehensive evaluation index of the HSS to be between 0 and 1.

### 3.3. Comprehensive Evaluation of the HSS in the BRI

#### 3.3.1. Characteristics of the HSS Index for the BRI Area

The evaluation of the HSS along the BRI regions shows that the HSS ranged from 0.07 to 1.00, with an average value of 0.66 (Figure 5). Spatially, the HSS gradually increased from the northeastern and southwestern parts of the study area to the central study area. The minimum HSS (<0.4) occurred in the Tibetan Plateau region. High-HSS (>0.9) regions were concentrated on the North China Plain, Middle–Lower Yangtze Plain, and Ganges Plain, as well as along the banks of the Nile and the coastal areas of the Mediterranean Sea.

The HSS for the vast East European Plain, West Siberian Plain, Inner Mongolian Plateau, and Northeast China Plain was relatively high (approximately 0.7). Except for the Inner Mongolia Plateau, all the regions with a relatively high HSS were plains, which had notable terrain advantages. The cold climate exerted a relatively significant negative impact on Central and Eastern Europe; however, this impact was corrected by the abundant surface water and vegetation resources in this region. As a result, the HSS for Central and Eastern Europe was still relatively high. Although the Inner Mongolian Plateau has a plateau terrain, the region consists mostly of plateau tablelands that differ relatively insignificantly in height. The HSS values have been positively improved by ground cover conditions in the plateau meadows; as a result, the HSS for this region was also relatively high. The HSS for the coastal regions of the Indochinese Peninsula, as well as Sumatra and Kalimantan (islands within the Malay Archipelago) in Southeast Asia, were also above 0.7. Low-HSS (approximately 0.5) regions were primarily distributed in the Rub’ al Khali Desert in West Asia and the Middle East, as well as on the Deccan Plateau in South Asia, the northern, mountainous Indochinese Peninsula, the Central Siberian Plateau, and the East Siberian Plateau. These regions have towering, complex terrains and overly extreme temperature and humidity conditions. Clearly, the HSS evaluation results were in relatively good agreement with the common geographical knowledge.

As shown is that the correlations between the single-factor indices and the results calculated using the comprehensive HSS model (Figure 6), and there was a positive, gradually strengthening correlation between the HSS and NRDLS. As a result of the normalization of the minimum RDLS values in the model calculations, a small number of low-value intervals of the NRDLS and the HSS distribution are relatively irregular. Homoplastically, the low values of the NTHI were a combined result of alpine conditions and wet and hot climates. In other words, within the HSS interval of 0–0.2, the NRDLS and NTHI both ranged from 0.1 to 0.5, whereas the positive impact of the LCI on the HSS was relatively insignificant. The LSWAI corrected the leading factors to a large extent. Based on the geographic spatial distribution characteristics, low-NRDLS regions were primarily located on the Tibetan Plateau and northern mountainous of Southeast Asia. These regions consist mainly of alpine meadows and deserts but also have relatively abundant surface water resources. Low-NTHI regions were located on the Alpine Tibetan Plateau, as well as the vast regions extending from Mongolia to Russia. 

As the HSS increased from 0.2 to 0.4, there was an increase in both the NRDLS and the NTHI, with some low values of the LSWAI (e.g., for the Rub’ al Khali Desert and the Deccan Plateau), and there was only a slight change in the LCI. As the HSS increased from 0.4 to 0.6, the NRDLS and the NTHI increased and decreased alternately, and the NRDLS exerted a relatively significant positive impact on the HSS. Additionally, as the HSS increased from 0.4 to 0.6, the corrective effect of the LSWAI and the LCI began to be enhanced. Specifically, the mountainous region of Southeast Asia had a relatively high land cover and abundant surface water resources. As the HSS increased beyond 0.6, there was almost no change in the NRDLS, but the LSWAI increased in a fluctuating manner. The increase in the HSS was a combined result of the linear increase in the THI and the correction by the LCI. This phenomenon was observed in low-RDLS regions, such as the East European Plain and the North China Plain. As the HSS increased beyond 0.9, the LCI increased sharply to above 0.3. The increase in the HSS was a result of the correction by the hydrological and land cover conditions.

#### 3.3.2. Zoning for Suitability Based on HSS in the BRI

Based on the terrain and climate suitability evaluation results, as well as the spatial distribution characteristics of the LSWAI and LCI, the HSS thresholds were determined. There was a trade-off relationship between the contribution of the RDLS and THI to the HSS. The HSS was limited by the two leading factors (i.e., terrain and climate). The following four cases were observed: the HSS was limited by two factors; the HSS was limited by a single factor, and there was a gradual increase in the single-factor suitability; and two factors were suitable (Table 3). When the HSS <0.61, the human settlement was limited by a single factor throughout and was critically unsuitable. When 0.61 < HSS > 0.7, there was an improvement in the single human settlement factors and an increase in the amount of surface water and land coverage, as the human settlement was critically suitable. In summary, the HSS in the study area was classified into six levels: namely, unsuitable (0 < HSS < 0.41), critically unsuitable (0.41 < HSS < 0.61), critically suitable (0.61 < HSS < 0.7), generally suitable (0.7 < HSS < 0.78), moderately suitable (0.78 < HSS < 0.88), and highly suitable (HSS > 0.88).

The HSS-based suitability zoning evaluation results showed that (Figure 7) human settlements that were found to be dominated by suitable conditions accounted for 63% of the study area. Spatially, unsuitable human settlements were primarily distributed on the Tibetan Plateau but also spread northeast and southwest from the Tibetan Plateau [17]. The HSS transitioned to the critically unsuitable level on the Deccan Plateau in India, the Mongolian Plateau, and the vast Siberian Plateau, as well as in the Rub’ al Khali Desert in West Asia. The HSS gradually increased in the region extending southward from the Siberian Plateau. Human settlements on the Northern West Siberian Plain and Southern Central Siberian Plateau, as well as in Daxing’anling Prefecture in Northeastern China, were found to provide critically suitable settlement conditions. Additionally, human settlements on the vast East European Plain, Southern West Siberian Plain, and Eastern Northeast China plain were found to provide generally suitable conditions. Human settlements that provided highly suitable conditions accounted for 6% of the study area and were mainly located in the North China Plain, Middle–Lower Yangtze Plain, and Ganges Plain, as well as along the banks of the Nile and coastal areas of the Mediterranean Sea. The HSS gradually decreased in the region extending from Southwestern China to Southeast Asia. In this region, only human settlements in Vietnam provided highly to moderately suitable conditions. As a result of the hot and humid climate, the human settlements in most of Southeast Asia provided relatively or generally suitable conditions. Furthermore, the human settlements that provided moderately and highly suitable conditions were located around the Nile Delta, the Asia Minor Peninsula, and the banks of the Caspian and Black Seas in West Asia.

## 4. Discussion

The BRI covers a vast area, and the Silk Road countries gifted with different resource endowments show strong economic complementarities as a whole unit [24], which leads to significant potential and a high scope for cooperation [10]. Moreover, the Silk Road countries are largely located in the areas sensitive to global climate change [20]. These have complex natural environments and diverse ecosystems [25]. Thus, it is of great practical significance to explore the humans settlements in these areas in order to understand the state of their natural resources and, finally, to achieve their sustainable development [3]. At present, the economy has continued to develop steadily in the regions along the Belt and Road, and the increasingly frequent human activities have had a more profound impact on the regional climate and ecological environment [23]. Among them, the changes in regional land cover and vegetation–hydrological conditions are particularly prominent [20]. Therefore, in response to changes in the natural environment elements, carrying out research on the human settlements in the BRI will help promote the rational distribution and orderly development of the region’s population [17]. 

The human settlement index model established in this paper based on the four physical and geographical factor indices is an exploration of the evaluation of the human settlements environment in the geographical region of the BRI. The evaluation process includes a logical analysis and correction of the data in combination with the actual situation. The results showed that each pairing of the four single-factor indices constituting the HSS model for the study area were subjected to a correlation analysis. In total, six correlation analyses were conducted. An extremely weak correlation was found between the RDLS and THI, such as the LSWAI and LCI. Thus, it was established with the RDLS and THI as the leading factors and the LSWAI and LCI as the auxiliary factors to correct the leading factors. The model was used to characterize the regional HSS as determined by terrain and climatic suitability and its enhancement by the LCI and LSWAI conditions. Based on the existing research and data correction, the HSS model is more scientific, but the evaluation results for different regions need to be further tested.

The evaluation results of the human settlements showed that topographic and climatic conditions are important limiting factors for the suitability of human settlements [26]. Due to the renewability and manmade adjustability of the hydrological and land cover conditions [19], the comprehensive suitability of the human settlements showed differences in different geographic spaces along the BRI [27]. In areas with relatively abundant surface water resources restricted by alpine conditions [28], it is recommended to increase the availability of the surface water in the summer and improve the coverage of the surface vegetation by improving the irrigation methods and technologies [29]. At the same time, the evaluation results of human settlements are expected to provide a decision-making basis for promoting population redevelopment and protecting ecosystems [20].

For a long period of time, compared with the two rigid limiting factors of the terrain (high altitude) and climate (cold), hydrology and land cover have been important breakthroughs in improving the suitability of human settlements [30]. According to the results of the suitability and restricted zoning of human settlements along the BRI region, water conservancy projects and technical measures (such as drip irrigation) should be used to improve seasonal water shortages and improve land cover conditions, so as to improve the suitability of the regional human settlements, making the best use of the situation and guiding the people in an orderly manner according to the time and place to gradually flow and migrate from areas with unsuitable human settlements to areas with critically suitable and suitable human settlements. Based on the evaluation results of the human settlements, it is possible to coordinate urban and rural development according to the local conditions, reasonably control the scale of cities and towns, reasonably increase residential settlements, guide the orderly development of the population, and promote the adaptation of population distribution to the suitability of the human settlement environment [31].

## 5. Conclusions

The physical and geographical conditions varied relatively significantly between the BRI regions, which are mainly affected by altitude and climate. For example, the topography, climate, and ground cover conditions of alpine regions such as the Tibetan Plateau restrict settlement, while the vast plains have a pleasant climate and rich land and water resources. In particular, the RDLS ranged from 0 to 19.9, with an average value of 1.05, with the value gradually decreased from the Himalayas and Tianshan Mountains to the surrounding regions. As for the THI, except for the Tibetan Plateau, the THI gradually decreased from the low-latitude regions in the south to the high-latitude regions in the north. The average LSWAI was approximately 0.33, and those values were clustered in space for the BRI regions. The multi-year average LCI was approximately 0.21. Low LSWAI and LCI regions were primarily concentrated in two types of zones—namely, alpine zones, as well as hot and dry zones.

This study developed a 1-km grid-based model of a comprehensive evaluation for HSS (based on the RDLS, THI, LSWAI, and LCI). The natural suitability and limitations of the potential human settlements in various countries and regions along the BRI were briefly evaluated. Based on the single-factor correlation analyses, the HSS model consists of a square formed by two 2D quadrants interlaced with the dominant factor as the coordinate origin, with a side length of 1.00. The NRDLS and NTHI, which were two extremely weakly correlated factors, were placed at diagonal corners of the square as the leading factors, and the LSWAI and LCI were used as the endpoint of the coordinates as the auxiliary factors. Thus, the HSS model consisted of two evaluation results—namely, the terrain and climatic suitability corrected by the hydrological and terrain indices. 

The HSS evaluation results were obtained based on the HSS model for the BRI area and revealed that the HSS for the BRI area ranged from 0.07 to 1.00, with an average value of 0.66. Spatially, the HSS gradually increased from the northeastern and southwestern parts of the study area to the central study area. Based on the terrain and climate suitability evaluation results, as well as the spatial distribution characteristics of the LSWAI and LCI, the HSS in the BRI was classified into six levels: namely, unsuitable, critically unsuitable, critically suitable, generally suitable, moderately suitable, and highly suitable. In particular, the human settlement suitability is dominated by critically suitable and suitable conditions, accounting for 63% of the BRI. Based on the above comprehensive zoning of the natural suitability of human settlements, it provides a basis for the rational distribution of the population and spatial planning in the BRI. The results of the study can be used to guide the migration of populations with unsuitable regions to critically suitable regions, as well as continuously improve the hydrology and land cover conditions in critically suitable regions, and maintain the natural ecological environment in suitable zones for human settlements, so as to realize the healthy life of human beings and the sustainable development of society.

## Figures and Tables

**Figure 1 ijerph-19-06044-f001:**
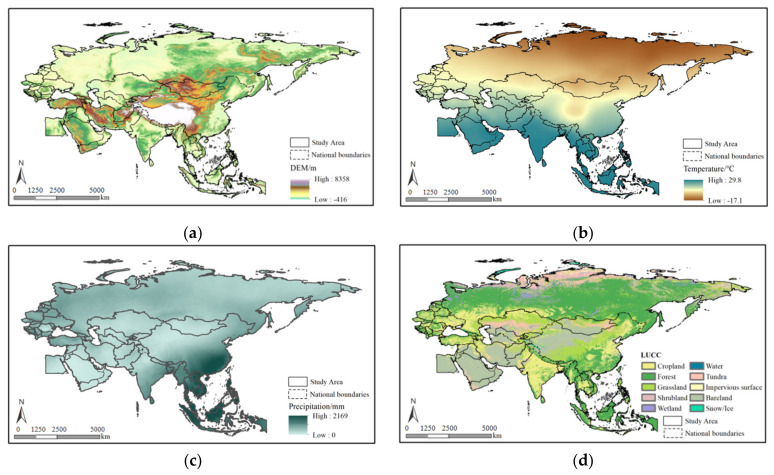
General details of the BRI area. Note: (**a**) the DEM of the BRI, (**b**) the mean annual temperature of the BRI, (**c**) the LSWI of the BRI, and (**d**) the LUCC of the BRI.

**Figure 2 ijerph-19-06044-f002:**
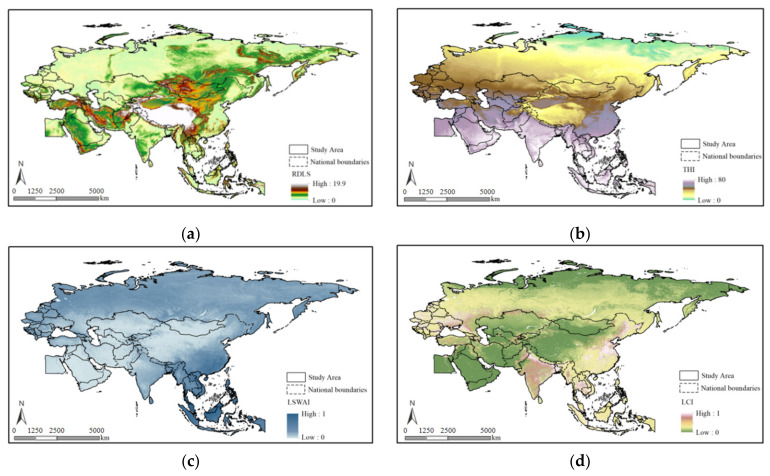
Spatial distribution of the single factors affecting human settlement suitability. Note: (**a**) the RDLS of the BRI, (**b**) the THI of the BRI, (**c**) the LSWAI of the BRI, and (**d**) the LCI of the BRI.

**Figure 3 ijerph-19-06044-f003:**
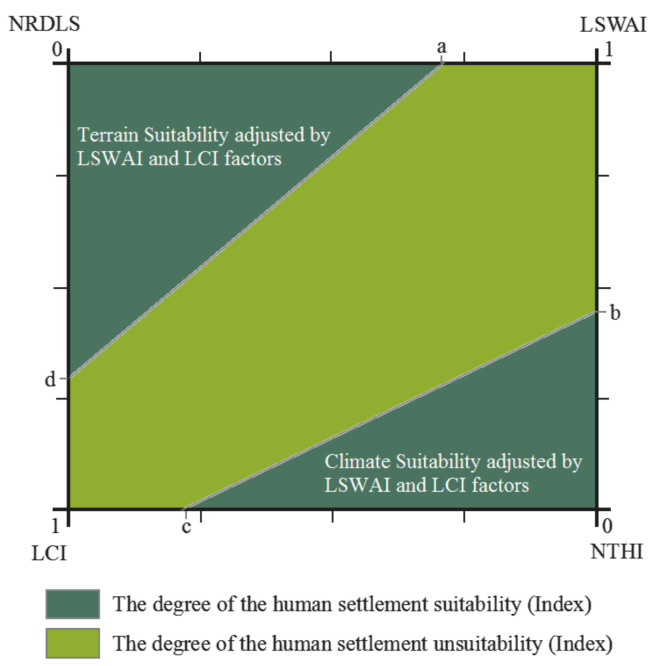
Schematic diagram of the spatial HSS model.

**Figure 4 ijerph-19-06044-f004:**
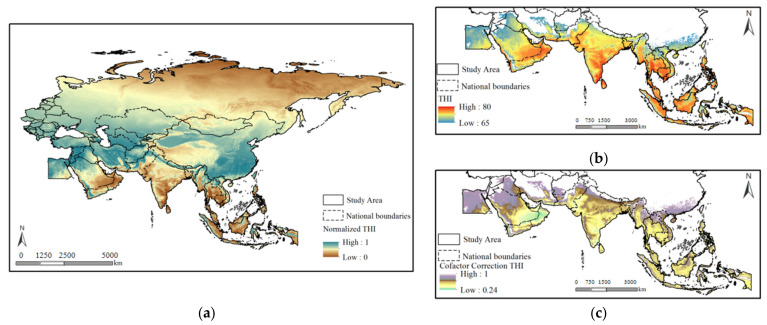
Piecewise corrected NTHI values for the BRI area. Note: (**a**) the normalized THI of the BRI, (**b**) the original THI of the BRI, and (**c**) the cofactor correction THI of the BRI.

**Figure 5 ijerph-19-06044-f005:**
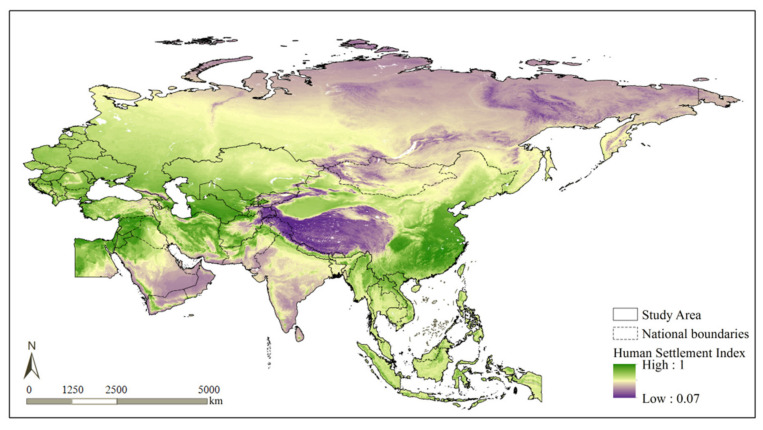
HSS model-based evaluation results.

**Figure 6 ijerph-19-06044-f006:**
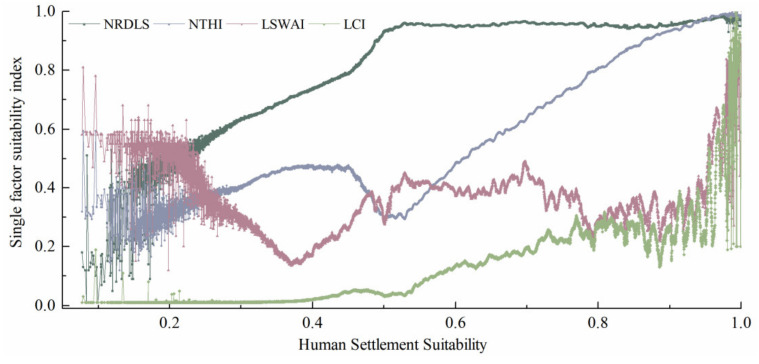
Correlations between the HSS and the single-factor index models.

**Figure 7 ijerph-19-06044-f007:**
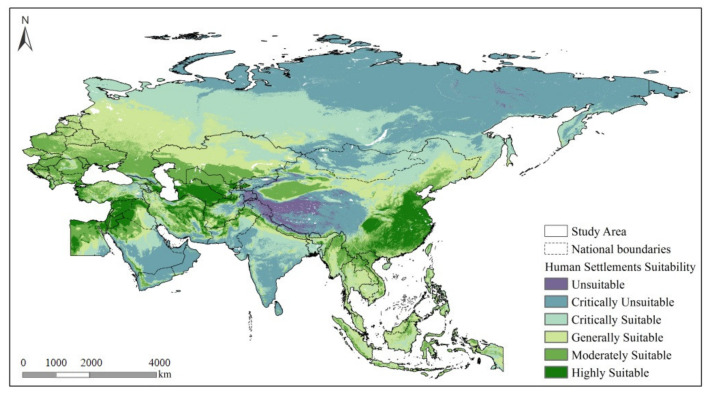
HSS-based zoning.

**Table 1 ijerph-19-06044-t001:** (**a**) Classification of the single-factor suitability indices—terrain suitability. (**b**) Classification of the single-factor suitability indices—climatic suitability.

**(a)**					
**Terrain Suitability**	**RDLS**	**Normalized RDLS (NRDLS)**	**Altitude**	**Relative Height Difference**	**Landform**
Highly suitable	0~0.2	0.97~1	<120	<50	Plains
Moderately suitable	0.2~0.4	0.96~0.97	120~180	40~110	Hills
Generally suitable	0.4~1.3	0.92~0.96	180~530	110~395	Mountains
Critically suitable	1.3~4.2	0.77~0.92	530~3500	37~535	Mountains and plateaus
Unsuitable	4.2~6.5	0.66~0.77	3500~5300	219~620	High mountains
Uninhabited	>6.5	<0.66	>5300	>620	Extremely high mountains
**(b)**					
**Climatic Suitability**		**THI**	**Somatic Sensation**		
Highly suitable		60~70	Warm and cool, and very comfortable		
Moderately suitable		50~60	Fresh and comfortable		
Generally suitable		45~50, 70~75	Relatively cold (hot) and relatively comfortable		
Critically suitable		40~45, 75~80	Cold (hot and humid) and uncomfortable		
Unsuitable		30~40, >80	Cold (extremely hot and humid) and uncomfortable		
Uninhabited		<30	Extremely cold and extremely uncomfortable		

**Table 2 ijerph-19-06044-t002:** Correlation coefficient matrix of the single HSS factors.

Correlation Coefficient	RDLS	THI	LSWAI	LCI
RDLS		0.155	0.186	0.244
THI			0.422	0.657
LSWAI				0.047
Extremely strongly correlated	Strongly correlated	Moderately correlated	Weakly correlated	Extremely weakly correlated
0.8–1.0	0.6–0.8	0.4–0.6	0.2–0.4	0.0–0.2

**Table 3 ijerph-19-06044-t003:** Correlation coefficient matrix of single factors for the HSS.

Threshold Determination	HSS	NRDLS	NTHI	LSWAI	LCI	Physical Meaning
Limited by the NRDLS and NTHI	0.35	0.68	0.45	0.20	0.01	Extremely high mountains, cold, lack of surface water, and low surface coverage
	0.41	0.75	0.47	0.21	0.03	Plateaus, cold, lack of surface water, and low surface coverage
	0.45	0.79	0.47	0.26	0.05	Mountains, cold, lack of surface water, and low surface coverage
Limited by the NRDLS or NTHI	0.61	0.95	0.50	0.37	0.14	Hills, cold, lack of surface water, and relatively low surface coverage
Gradual increase in the NRDLS or NTHI	0.70	0.96	0.64	0.47	0.19	Plains and hills, cold (hot and humid), relatively abundant surface water, and relatively low surface coverage
Gradual increase in the NRDLS or NTHI	0.74	0.96	0.71	0.33	0.21	Plains and hills, relatively cold (hot), relatively abundant surface water, and relatively high surface coverage
Gradual increase in the NRDLS or NTHI	0.78	0.95	0.78	0.29	0.25	Plains and hills, cool, relatively abundant surface water, and relatively high surface coverage
Two factors are suitable, as well improved by the NRDLS and NTHI	0.88	0.95	0.92	0.37	0.23	Plains and hills, warm, relatively abundant surface water, and relatively high surface coverage
	0.97	0.98	0.99	0.56	0.48	Plains, warm, abundant surface water, and high surface coverage
Single-factor suitability level	Unsuitable	Critically suitable	Generally suitable	Moderately suitable	Highly suitable	

## Data Availability

Some or all data and models that support the findings of this study are available from the corresponding author upon reasonable request.

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
