# Peer review of "GIS-Based Modeling of Human Settlement Suitability for the Belt and Road Regions"

_ijerph, 2022, doi:10.3390/ijerph19106044_

Round 1

Reviewer 1 Report

  1. The paper evaluates the human settlements suitability of the countries and regions along the Belt and Road through the construction of geographic grid. The research method has innovative value, and the data basis is detailed, and the results are thoroughly analyzed. However, the development strategies proposed on the basis of the research results are relatively weak.
  2. The review of the existing research on human settlement environment suitability is too simple and there are few references;
  3. The paper selected RDLS, THI, LSWAI and LCI indicators for correlation analysis and evaluation model construction, but did not elaborate on the reasons for selecting these indicators.
  4. The article briefly presents the importance and impact of human settlements suitability, but does not evaluate or discuss relevant research. For example, the methodology used in this study in L.47 is evolving and can be briefly described. So what are the improvements of this research method compared with the previous one?
  5. The various indicators selected for the study are explained in detail, but the reasons for their selection and the feasibility of their use in assessing human settlements suitability are not explained much.
  6. It is suggested to indicate the names of countries or regions with the highest and lowest values of each index in Figure 1-2 to improve the readability of the article.
  7. The significance of this study was emphasized in the discussion. However, it is more desirable to see the author put forward relevant strategies and suggestions to relevant government departments or professional fields according to the research results, and put forward future research prospects.
  8. Why are RDLS, THI, LSWAI and LCI selected and how are they considered? How can the accuracy and rationality of the comprehensive HSS evaluation model for the countries and 132 regions along the Belt and Road be verified?
  9. The drawings shall be expressed in a unified and standardized manner. "Legend" is missing in Figures 1, 2, 4 and 7.10. The discussion should be combined with the introduction, and the deficiencies of this study and the prospects of future research should be added.
  10. The paper uses GIS as the means to model the natural suitability of the belt and Road Region and tries to divide the suitability of the region by constructing an evaluation system. From the perspective of the theme of the paper, it is difficult and has a large workload, which has good academic research and reference value. However, the concept of "human settlements" does not exist in the paper, and as far as the actual paper is concerned, the scope of its research has gone beyond the scope represented by human settlements. At the same time, the evaluation of natural suitability is not a simple terrain, altitude, water and climate, but also includes many other factors. However, the paper does not enumerate and analyze the factors, and the influencing factors are weak.
  11. The paper takes the quality of human settlements as the research objective, and takes the countries surrounding the Belt and Road As the research object, covering a wide area with rich cultural and geographical conditions, which fully reflects the representativeness of research samples. At the same time the author also constructs the human settlemen t suitability (HSS) evaluation model, chose the surface fluctuation degree (RDLS), temperature humidity index THI, land surface water LSWAI abundance index, the surface coverage LCI objective index as the research basis, such as These indicators can well reflect people's living conditions. Compared with people's subjective feelings, objective indicators can better reflect the group as a unit of environmental livable level. However, the research significance of this paper is too broad and does not give reasonable and specific application scenarios. Simply pointing out that its research is conducive to sustainable development is of no practical value.
  12. The paper used the ASTERGDEM GIS Platform, NASA Earth Data Platform highly credible data providers, such as extracts such as climate datasets, Land – surface water index, Land-cover data and other rich data types, fully consider the time and space span of data, the workload is very rich. In addition, the author gives different weights to different data factors in data processing, so as to adjust the balance between data according to the different amount of data and prevent extreme results. Regularization of data has the same effect. In the HSS analysis method adopted by the author, the role of the four indicators in the human settlement environment has not been fully elaborated. In 3.3.1, the correlation between the four indicators and the suitability of the human settlement environment is given, but the evaluation standard of the environmental suitability is not given. In Figure 6, HSS model aims to reflect the relationship between a single factor and habitat suitability, but the causal relationship reflected in the diagram is opposite. If there is a special purpose also need to declare the author.
  13. In Chapter 3, a large number of charts are used for suitability analysis of each indicator, and data processing of human perception evaluation is conducted in the form of prior. The color model of different indexes in the research area is presented intuitively by using the map information reasonably. However, in Section 2.3, the author mentioned a large number of geographical location dimensions and altitude information, and made statistics on the distribution of mountain ranges, plains, rivers and other geomorphic features in different regions, which was irrelevant to data analysis and somewhat redundant.

Author Response

Reviewer 1

  1. The paper evaluates the human settlements suitability of the countries and regions along the Belt and Road through the construction of geographic grid. The research method has innovative value, and the data basis is detailed, and the results are thoroughly analyzed. However, the development strategies proposed on the basis of the research results are relatively weak.

Thanks to the reviewer for your affirmation of the research content, research methods, and analysis results of this paper. At the same time, for the relatively weak development strategy mentioned by the expert, corresponding supplements are made in the manuscript, see lines 499-513 for details. Manuscript revised to:

In a long period of time, compared with the two rigid limiting factors of terrain (high altitude) and climate (cold), hydrology and land cover are important breakthroughs in improving the suitability of human settlements. According to the results of the suitability and restricted zoning of human settlements along the BRI region, water conservancy projects and technical measures (such as drip irrigation) should be used to improve seasonal water shortages and improve land cover conditions, so as to improve the suitability of regional human settlements. At the same time, make the best use of the situation and guide the people in an orderly manner according to the time and place to gradually flow and migrate from areas with unsuitable human settlements to areas with critically suitable and suitable human settlements. Based on the evaluation results of human settlements, it is possible to coordinate urban and rural development according to local conditions, reasonably control the scale of cities and towns, reasonably increase residential settlements, guide the orderly development of the population, and promote the adaptation of population distribution to the suitability of the human settlements environment.

  1. The review of the existing research on human settlement environment suitability is too simple and there are few references;

Thanks to the expert for your professional suggestions, the research review has been further reviewed, and references have been supplemented.

See manuscript and references for details.

  1. The paper selected RDLS, THI, LSWAI and LCI indicators for correlation analysis and evaluation model construction, but did not elaborate on the reasons for selecting these indicators.

Thanks to the reviewer for your careful review and professional opinions, at the beginning of the model construction, supplementary explanations for the reasons for the selection of indicators. See lines 269-274 for details. Manuscript revised to:

There are various factors affecting the suitability of human settlements, but the most fundamental and determining factors of other natural environment, the main factors that play a leading role in the natural environment of human settlements mainly include the degree of terrain fluctuation, hydrothermal and climatic conditions, and hydrological conditions, as well as the characteristics of land use and vegetation coverage that comprehensively reflect the natural conditions of the region.

  1. The article briefly presents the importance and impact of human settlements suitability, but does not evaluate or discuss relevant research. For example, the methodology used in this study in L.47 is evolving and can be briefly described. So what are the improvements of this research method compared with the previous one?

Thanks to the reviewer for your professional opinions, based on the expert opinions, the research review will be further supplemented and discussed. See lines 49-52 for details. Manuscript revised to:

Existing HSS methods use a simple model of single-element grading and spatial superposition of single elements for comprehensive grading. Previous study methods did not consider the influence degree and interaction relationship of elements in the evaluation model

  1. The various indicators selected for the study are explained in detail, but the reasons for their selection and the feasibility of their use in assessing human settlements suitability are not explained much.

Thanks to the reviewer for your careful review and professional advice. The evaluation indicators of natural elements selected in this paper are relatively comprehensive, and the basic data are easy to obtain, which can objectively evaluate the suitability of large-scale human settlements. The reasons for the selection of each indicator have been supplemented in the revised draft, see lines 269-274 for details. Manuscript revised to:

“There are various factors affecting the suitability of human settlements, but the most fundamental and determining factors of other natural environment, the main factors that play a leading role in the natural environment of human settlements mainly include the degree of terrain fluctuation, hydrothermal and climatic conditions, and hydrological conditions, as well as the characteristics of land use and vegetation coverage that comprehensively reflect the natural conditions of the region.”

  1. It is suggested to indicate the names of countries or regions with the highest and lowest values of each index in Figure 1-2 to improve the readability of the article.

Thanks to the reviewer for your careful review. On the professional opinions of expert, a brief explanation is given after discussion by the research team. This paper selects the Belt and Road as the study area, and conducts zoning and evaluation of the suitability of large-scale human settlements based on physical and geographical elements. The evaluation results are based on the spatial description of the characteristics of human settlements by natural geographical divisions, which weakens the characteristics of human settlements in different administrative divisions, so as not to carry out an intuitive comparison of suitability characteristics between countries. We will supplement the differences in human settlement suitability between countries in future research.

  1. The significance of this study was emphasized in the discussion. However, it is more desirable to see the author put forward relevant strategies and suggestions to relevant government departments or professional fields according to the research results, and put forward future research prospects.

The professional advice of reviewer is of great significance for improving the height of the paper. Based on expert opinions, supplement relevant strategies and suggestions in the discussion section, see lines 499-513 for details. Manuscript revised to:

In a long period of time, compared with the two rigid limiting factors of terrain (high altitude) and climate (cold), hydrology and land cover are important breakthroughs in improving the suitability of human settlements. According to the results of the suitability and restricted zoning of human settlements along the BRI region, water conservancy projects and technical measures (such as drip irrigation) should be used to improve seasonal water shortages and improve land cover conditions, so as to improve the suitability of regional human settlements. At the same time, make the best use of the situation and guide the people in an orderly manner according to the time and place to gradually flow and migrate from areas with unsuitable human settlements to areas with critically suitable and suitable human settlements. Based on the evaluation results of human settlements, it is possible to coordinate urban and rural development according to local conditions, reasonably control the scale of cities and towns, reasonably increase residential settlements, guide the orderly development of the population, and promote the adaptation of population distribution to the suitability of the human settlements environment.

  1. Why are RDLS, THI, LSWAI and LCI selected and how are they considered? How can the accuracy and rationality of the comprehensive HSS evaluation model for the countries and 132 regions along the Belt and Road be verified?

I am very grateful to the expert for your careful review. The professional opinions of expert are of great significance for improving the height of the paper and the continuity of the research. As for the question“How can the accuracy and rationality of the comprehensive HSS evaluation model for the countries and 132 regions along the Belt and Road be verified?”,Continued and in-depth discussions will be made in future study, and expert opinions will be of great value to our team's future study directions.

The reasons for choosing the four indices are briefly explained. Manuscript revised to:

There are various factors affecting the suitability of human settlements, but the most fundamental and determining factors of other natural environment, the main factors that play a leading role in the natural environment of human settlements mainly include the degree of terrain fluctuation, hydrothermal and climatic conditions, and hydrological conditions, as well as the characteristics of land use and vegetation coverage that comprehensively reflect the natural conditions of the region. According to the degree of influence of naturally geographical elements on the human settlement environment, four types of constituent factors are selected. The index system has been used experimentally in China. The results demonstrated that the four basic factors (RDLS, THI, LSWAI, and LCI) affecting human settlements could effectively characterize their comprehensive suitability.

  1. The drawings shall be expressed in a unified and standardized manner. "Legend" is missing in Figures 1, 2, 4 and 7.10. The discussion should be combined with the introduction, and the deficiencies of this study and the prospects of future research should be added.

Thanks to the reviewer for your careful review, the "Legend" in Figure 5 has been removed due to the unnecessary cartographic elements and the unified cartographic specification.

And supplemented the discussion on the lack of research and future research prospects. Manuscript revised to:

Based on the existing research and data correction, the HSS model is more scientific, but the evaluation results for different regions need to be further tested.

  1. The paper uses GIS as the means to model the natural suitability of the belt and Road Region and tries to divide the suitability of the region by constructing an evaluation system. From the perspective of the theme of the paper, it is difficult and has a large workload, which has good academic research and reference value. However, the concept of "human settlements" does not exist in the paper, and as far as the actual paper is concerned, the scope of its research has gone beyond the scope represented by human settlements. At the same time, the evaluation of natural suitability is not a simple terrain, altitude, water and climate, but also includes many other factors. However, the paper does not enumerate and analyze the factors, and the influencing factors are weak.

I am very grateful to the reviewer for your affirmation of the research topic, research method, research content and workload. The opinions of the experts are the driving force for the research team to continue to carry out the evaluation of the suitability of human settlements.

Through careful discussion and literature reading by the research team, the concept and connotation of "human settlement" are briefly explained. "Human settlement" includes the aspect of choosing the place of residence based on the subjective will of people, the aspect of objectively determining or restricting the place of residence by environmental factors, and the aspect of improving the place of residence through economic and technological means. Among them, the objective determination of residence by environmental factors is mainly natural factors, that is, the natural suitability of human settlements discussed in this paper.

At the same time, the research team of this paper agrees with the reviewers' statement that "the evaluation of natural suitability is not a simple terrain, altitude, water and climate, but also includes many other factors", but also includes many factors such as extreme weather and sudden natural disasters. For large-scale grid evaluation, data availability and the applicability of the research scale have to be considered. Agreeing with the opinions put forward by expert, the research team will include more influencing factors in future research to enrich the connotation of the natural suitability of human settlements, and try to meet the accuracy requirements of small-scale human settlement suitability evaluation.

  1. The paper takes the quality of human settlements as the research objective, and takes the countries surrounding the Belt and Road As the research object, covering a wide area with rich cultural and geographical conditions, which fully reflects the representativeness of research samples. At the same time the author also constructs the human settlemen t suitability (HSS) evaluation model, chose the surface fluctuation degree (RDLS), temperature humidity index THI, land surface water LSWAI abundance index, the surface coverage LCI objective index as the research basis, such as These indicators can well reflect people's living conditions. Compared with people's subjective feelings, objective indicators can better reflect the group as a unit of environmental livable level. However, the research significance of this paper is too broad and does not give reasonable and specific application scenarios. Simply pointing out that its research is conducive to sustainable development is of no practical value.

I am very grateful to the reviewer for your affirmation of the research object, research scale, research method, indicator selection, research content, etc. The professional opinions and suggestions of experts are important references for improving the quality of the paper.

According to the opinions of expert, the problem of "research significance is too broad" is revised, and specific application scenarios are supplemented in the discussion section. Manuscript revised to:

In a long period of time, compared with the two rigid limiting factors of terrain (high altitude) and climate (cold), hydrology and land cover are important breakthroughs in improving the suitability of human settlements. According to the results of the suitability and restricted zoning of human settlements along the BRI region, water conservancy projects and technical measures (such as drip irrigation) should be used to improve seasonal water shortages and improve land cover conditions, so as to improve the suitability of regional human settlements. At the same time, make the best use of the situation and guide the people in an orderly manner according to the time and place to gradually flow and migrate from areas with unsuitable human settlements to areas with critically suitable and suitable human settlements. Based on the evaluation results of human settlements, it is possible to coordinate urban and rural development according to local conditions, reasonably control the scale of cities and towns, reasonably increase residential settlements, guide the orderly development of the population, and promote the adaptation of population distribution to the suitability of the human settlements environment.

  1. The paper used the ASTERGDEM GIS Platform, NASA Earth Data Platform highly credible data providers, such as extracts such as climate datasets, Land – surface water index, Land-cover data and other rich data types, fully consider the time and space span of data, the workload is very rich. In addition, the author gives different weights to different data factors in data processing, so as to adjust the balance between data according to the different amount of data and prevent extreme results. Regularization of data has the same effect. In the HSS analysis method adopted by the author, the role of the four indicators in the human settlement environment has not been fully elaborated. In 3.3.1, the correlation between the four indicators and the suitability of the human settlement environment is given, but the evaluation standard of the environmental suitability is not given. In Figure 6, HSS model aims to reflect the relationship between a single factor and habitat suitability, but the causal relationship reflected in the diagram is opposite. If there is a special purpose also need to declare the author.

I am very grateful to the reviewer for your affirmation of the selection of data sources, data processing and workload in this paper, and correct the problems mentioned by the experts one by one in the corresponding positions in the text, specifically:

  • The role of the four indicators in the human settlement environment.

Manuscript revised to: 

The previous correlation analyses determined an extremely weak correlation between the NRDLS and the NTHI, so as the LSWAI and LCI. Thus, the weakly correlated factors were “independent” but not “influential”. Therefore, it was necessary to place the factors at “adjacent” locations that had a moderate or stronger correlation with one another, when constructing the comprehensive HSS evaluation model. To take advantage of the correlations between the auxiliary and leading factors that affect or enhance the HSS. Based on this logic, a spatial model and an index model were constructed in this study to evaluate HSS.

Based on the correlations between the four factors (i.e., the normalized RDLS (NRDLS), normalized THI (NTHI), LSWAI, and LCI), the NRDLS and NTHI were determined to be leading factors, and the LSWAI and LCI were considered to be auxiliary factors. The auxiliary factors were used to enhance the comprehensive HSS model determined by the leading factors. Based on this logic, spatial and index models were established separately.

  • The suitability evaluation criteria have been described in the text.Manuscript revised to:

the HSS in the study area was classified into six levels, namely, unsuitable (0< HSS <0.41), critically unsuitable (0.41< HSS <0.61), critically suitable (0.61< HSS <0.7), generally suitable (0.7< HSS <0.78), moderately suitable (0.78< HSS <0.88), and highly suitable (HSS >0.88).

  • Thanks to the reviewer for your careful review, the data and expressions are indeed the causal relationship between the single factor and the comprehensive HSS, that is, with the change of the comprehensive index, how does the single factor change to characterize the contribution of the single factor index to the comprehensive index of human settlements. The original data and software screenshots are attached:

  1. In Chapter 3, a large number of charts are used for suitability analysis of each indicator, and data processing of human perception evaluation is conducted in the form of prior. The color model of different indexes in the research area is presented intuitively by using the map information reasonably. However, in Section 2.3, the author mentioned a large number of geographical location dimensions and altitude information, and made statistics on the distribution of mountain ranges, plains, rivers and other geomorphic features in different regions, which was irrelevant to data analysis and somewhat redundant.

I am very grateful to the reviewer for your affirmation of the data processing and map presentation in this paper.

As mentioned in the above comments, this paper constructs the HSS model through RDLS, THI, LSWAI and LCI. The four index models are applied to basic data such as altitude, temperature, relative humidity, LSWI, rainfall, LUCC, NDVI, etc. Therefore, a basic geographic element is selected for each index to summarize the natural conditions of the study area.

The redundancies in the expression mentioned by the reviewer have been appropriately deleted in the paper.

Thank you for your efforts to improve our manuscript. We hope that our responses and the resulting changes will be acceptable, but we will be happy to work with you to resolve any remaining issues.

Reviewer 2 Report

This manuscript proposes a grid-based human settlements suitability (HSS) model to evaluate he natural suitability of human settlements. The topic is interesting, and this model provides a viewpoint in terms of global cooperation within sustainable aspect. I list my suggestions and comments to be addressed for reaching the requirement of IJERPH.

1) The study area—the Belt and Road Regions is the main contribution of this paper However, in the introduction the authors might want to provides more discussions on the specifications of this region. For example, why this HSS evaluation model is suitable for the Belt and Road Regions, what is the difference if this HSS was used in other regions?

2) The authors mentioned that the population distribution pattern could affect the natural environment of human settlements suitability. However, this factor is not considered in the proposed HSS evaluation model.

3) Some errors can be found in the captions of figures and content of figures. E.g. “Figure 1. General details of the BRI area. Note: (a) the DEM of the BRI, (b) the mean annual temperature of the BRI, (b) the LSWI of the BRI, and (b) the LUCC of the BRI.” (c) and (d) are mistakenly written as (b) and (b).

4) To my understanding, all dataset listed in the Subsection “Materials” are the existing ones. If so, all dataset should be visualized in a figure.

5) The results shown in Figure 2 seems not relevant. According to the section title “Single-factor index analysis of human settlements in the BRI”, these results should be the HSS results evaluated by various single factors. Similar issues are also in the results of Table 2.

6) Is the Piecewise corrected NTHI values was proposed by this manuscript? If so, the authors might want to explain why these thresh values were used in Equation (10) - (12). Or, the result of THI should not be mentioned separately, because the highlight of this manuscript is “comprehensive”, rather than “improving a single factor”.

7) The authors didn’t mention the information regarding GIS modeling method. According to the manuscript, the GIS method is kriging interpolation. If so, I don’t think GIS-based modeling could not be appropriate for the paper title. Moreover, the authors should provide which GIS platform was used in this paper.

8) I can accept the reasons for the result shown in Figure 7. It would be better if the authors can use some auxiliary data (e.g. overlapping with urban areas) to make this result more valid.

9) Typos and grammars could be found in the section title, captions, figure content, and the body of manuscript. The authors should polish the language thoroughly.

Author Response

Reviewer 2

This manuscript proposes a grid-based human settlements suitability (HSS) model to evaluate he natural suitability of human settlements. The topic is interesting, and this model provides a viewpoint in terms of global cooperation within sustainable aspect. I list my suggestions and comments to be addressed for reaching the requirement of IJERPH.

Thanks to the reviewer for your affirmation of research topics and research methods. Your affirmation is the driving force for the research team to continue to carry out research in this direction. In response to the comments made by the reviewer, the authors of this paper will coordinate and revise one by one carefully to meet the publication requirements.

  • The study area—the Belt and Road Regions is the main contribution of this paper However, in the introduction the authors might want to provides more discussions on the specifications of this region. For example, why this HSS evaluation model is suitable for the Belt and Road Regions, what is the difference if this HSS was used in other regions?

Some clarifications need to be made about the model used in this paper. Since 2008, the author's research team has pioneered the evaluation model of the suitability of human settlements based on physical and geographical elements. This paper is a new model based on the existing research of the research team, and it is the first application in the Belt and Road region.

Strongly agree with the reviewer's statement that the regional limitations of model application need to be clarified. The research team in this paper will conduct in-depth research on the differences in the application of HSS models in different regions in future research. At the same time, supplementary descriptions of expert opinions are provided. Manuscript revised to:

Based on the existing research and data correction, the HSS model is more scientific, but the evaluation results for different regions need to be further tested.

  • The authors mentioned that the population distribution pattern could affect the natural environment of human settlements suitability. However, this factor is not considered in the proposed HSS evaluation model.

Thanks to the reviewer for your careful review. The research team fully absorbs the professional opinions of reviewer, and will consider the relationship between population distribution and the suitability of human settlements in future research. The opinions of experts provide directions for future research.

Looking for manuscripts related to the representation for “population distribution”,It is inferred that the statement stated by the reviewer is: “The natural environment of human settlements exerts a significant impact on the development of human society, to a certain extent, dictates the spatial population distribution pattern”.

The above statement is intended to express the human settlement as the cause, and the population distribution as the result, that is, the population distribution is affected by the conditions of the human settlement. In other words, the evaluation of the suitability of human settlements can be used as the basis for the population distribution pattern. It is an application of the results of this paper, not the data used in this paper.

  • Some errors can be found in the captions of figures and content of figures. E.g. “Figure 1. General details of the BRI area. Note: (a) the DEM of the BRI, (b) the mean annual temperature of the BRI, (b) the LSWI of the BRI, and (b) the LUCC of the BRI.” (c) and (d) are mistakenly written as (b) and (b).

I am very grateful to the reviewer for your careful review. There are some unnecessary errors in the manuscript, which have been corrected. Manuscript revised to:

Figure 1. General details of the BRI area. Note: (a) the DEM of the BRI, (b) the mean annual temperature of the BRI, (c) the LSWI of the BRI, and (d) the LUCC of the BRI.

  • To my understanding, all dataset listed in the Subsection “Materials” are the existing ones. If so, all dataset should be visualized in a figure.

Thanks to the expert for your careful review, the data, temperature, relative humidity and average annual rainfall described in the article are obtained through spatial interpolation through the GIS platform, and other data are existing datasets. Considering that there is a lot of data involved in this paper, a total of 7 basic data sets that make up 4 indices are involved. In order to ensure the beauty of the plot and the simplicity of the overview of the study area, a representative basic data is selected for each index to be presented. Thus, no visualization was performed for conventional NDVI and LSWI datasets, etc.

  • The results shown in Figure 2 seems not relevant. According to the section title “Single-factor index analysis of human settlements in the BRI”, these results should be the HSS results evaluated by various single factors. Similar issues are also in the results of Table 2.

I am very grateful to the reviewer for your careful review. For the problems mentioned by the expert, the author makes a few explanations.

The HSS described in this paper is derived from further model calculations with 4 single-factor indices, including RDLS, THI, LSWAI, and LCI. Therefore, before constructing and describing the HSS, it is necessary to express the four single-factor indices in the study area, namely RDLS, THI, LSWAI and LCI.

Without avoiding ambiguity, it is expressed in the abstract. Manuscript revised to:

Methods: This study presents a kilometer grid-based comprehensive human settlement suitability (HSS) evaluation model (containing the relief degree of land surface (RDLS), the temperature humidity index (THI), the land surface water abundance index (LSWAI), and the land-cover index (LCI)).

  • Is the Piecewise corrected NTHI values was proposed by this manuscript? If so, the authors might want to explain why these thresh values were used in Equation (10) - (12). Or, the result of THI should not be mentioned separately, because the highlight of this manuscript is “comprehensive”, rather than “improving a single factor”.

The paper does make a segmentation correction for NTHI. This section is described for three reasons: First, the smaller the value of RDLS, the better the terrain conditions, the larger the LSWAI and LCI values, the better the hydrological and land cover conditions, while the THI is different, the middle value indicates the best, and the maximum and minimum values indicate general climatic conditions. Therefore the results of NTHI are mentioned separately. Second, this problem was indeed found in the data processing process, and NTHI as a dominant factor had an important impact on the comprehensive suitability evaluation results. Third, the interpretation of the threshold in formulas (10) - (12) is briefly described in the paper. Manuscript revised to:

Therefore, smoothing based on the same interval threshold, as well as the relevant equations was required. For example, the THI values of 45-50 and 70-75 belong to the types of generally suitable climate. Based on the above, the data whose THI value is greater than 65 is processed in segments.

  • The authors didn’t mention the information regarding GIS modeling method. According to the manuscript, the GIS method is kriging interpolation. If so, I don’t think GIS-based modeling could not be appropriate for the paper title. Moreover, the authors should provide which GIS platform was used in this paper.

Thanks to the reviewer for your professional comments. All the data used in this paper are raster data, and the calculation and construction of all indices are based on the GIS platform. In addition, the statistical information described in this paper is also obtained through the spatial analysis module and operation of the GIS platform. As for the GIS platform pointed out by expert, it is obtained by comprehensively using the spatial analysis module and grid operation of ArcGIS 10.5 software. For the above reasons, the term "GIS-based" is included in the title.

  • I can accept the reasons for the result shown in Figure 7. It would be better if the authors can use some auxiliary data (e.g. overlapping with urban areas) to make this result more valid.

Many thanks for expert advice. Different base map elements were considered when drawing the map, and an attempt was made to superimpose the urban boundaries. But for the large-scale study area of the Belt and Road, the vector base map will hide more information. The author will fully absorb the opinions of expert, and will superimpose more base map elements in the future research on the suitability of small-scale human settlements.

  • Typos and grammars could be found in the section title, captions, figure content, and the body of manuscript. The authors should polish the language thoroughly.

Many thanks to the expert for your careful review, the authors will coordinately revise the full text and polish the language.

Thank you again for your efforts to improve our manuscript. We hope that our responses and the resulting changes will be acceptable, but we will be happy to work with you to resolve any remaining issues.

Reviewer 3 Report

This research developed an HSS model and analyzed the settlement suitability in Belt and Road Regions, where have a vulnerable environment and is promising in development in the near future. The topic is in line with the journal International Journal of Environmental Research and Public Health, especially very suitable for the special issue Ecosystem Health and Environmental Geography in the Belt and Road Regions. The results will make a good contribution to sustainable development planning in relevant areas and would be interesting to readers of the journal. Here are some suggestions:

In the introduction section:

  • Please refine the logic of the introduction section. for instance, lines 66 to 72, it's more like the description in the methods part. Revise the relevant parts to help readers get what you want to express, and present the background of why we need to answer the questions you pointed out in lines 87 to 95 at present.
  • It is better to supply the state of the art about the human settlements in BRI region and the relevant studies on this topic.
  • Line 73, when it comes to HSS, what is the main structure of HSS and how is it practically applied at present? This information is fundamental but missing.

In the Materials and Methods section,

  • Line 108, considering the widespread of your research area, it would be better to point out what is the quantity of the stations, do these stations spread evenly or satisfied for the interpolation resolution of this research?
  • Lines 132 to 134, why you choose these four indices? you can add some explanations in the introduction part.

In the discussion section:

  • Relevant research has been cited but not discussed. Please add some key points of these research and some analysis to address the importance and meaning of your research results.
  • the recommendations, lines 472-474, seem cannot be well supported by the results in this research. Please supply more explanations.

Conclusion:

  • Please revise and refine your conclusion section to better answer the scientific questions you addressed at the beginning of the manuscript.

Author Response

Reviewer 3

This research developed an HSS model and analyzed the settlement suitability in Belt and Road Regions, where have a vulnerable environment and is promising in development in the near future. The topic is in line with the journal International Journal of Environmental Research and Public Health, especially very suitable for the special issue Ecosystem Health and Environmental Geography in the Belt and Road Regions. The results will make a good contribution to sustainable development planning in relevant areas and would be interesting to readers of the journal. Here are some suggestions:

I am very grateful to the reviewer for your affirmation of the research topic, the fit of the topic, and the research results. The authors will make careful revisions one by one according to the opinions of reviewer.

In the introduction section:

  • Please refine the logic of the introduction section. for instance, lines 66 to 72, it's more like the description in the methods part. Revise the relevant parts to help readers get what you want to express, and present the background of why we need to answer the questions you pointed out in lines 87 to 95 at present.

I am very grateful for the professional opinion of the expert, according to which the preface has been readjusted.

In order to respond to the scientific problems to be solved in this paper (lines 87-95), the statement in the preface is revised, specifically:

  • Find out the natural background conditions of the Belt and Road regions, such as topography, climate, hydrology and land cover.

There is a large geographical area along the BRI routes. The countries along the BRI routes have different natural resources and are relatively highly economically developed. There is a great potential for cooperation between these countries. Additionally, the regions involved in the BRI are situated in a zone sensitive to climate change, with complex natural conditions and diverse and vulnerable natural environments. [Line 61-66]

  • According to the degree of influence of physical and geographical factors in the BRI regions for the human settlement environment, and the correlation between the factors, this study built a human settlement suitability index model.

This study is based on the 17 goals of the United Nations 2030 Agenda for Sustainable Development, with particular emphasis on meeting everyone's life needs in terms of health, safety, and sanitation. When natural conditions cannot satisfy everyone in terms of climate environment, water use, food security, etc., it is necessary to strengthen cooperation and promote technological innovation to improve the limitation of settlement environment caused by the natural geographical factors. Therefore, before this, it is necessary to clarify the degree of influence of different natural and geographical factors on the human settlements environment, and to improve the settlements environment of local residents in a targeted manner according to local conditions, so as to achieve the goal of allowing everyone to live a healthy life.   [Modify the lines 66-72]

  • Based on the GIS platform, form a raster atlas for comprehensive evaluation of human settlements in the BRI regions, and conduct grading and zonal evaluation of the human settlements in the study area, providing a research basis for the rational distribution of the population along the Belt and Road.

On the one hand, a more objective basis is needed to determine the key critical thresholds for classifying suitability (unsuitable, critically suitable, generally suitable, moderately suitable, and highly suitable) in HSS evaluation study.  [Line 95-99]

  • It is better to supply the state of the art about the human settlements in BRI region and the relevant studies on this topic.

I am very grateful to the reviewer for your careful review. The evaluation of the suitability of human settlements is the result of years of accumulation and continuous research by the author's research team. The scope of its research is concentrated on the county scale in China, and no relevant topics have been found for large-scale human settlements. According to expert opinions, the research progress of human settlement environment is supplemented.

Manuscript revised to:

The study development of human settlements has gradually transitioned from the perspective of architecture and urban planning to the perspective of geography, and combined with 3S technology to establish a grid-scale evaluation index for the natural suitability of human settlements. Among them, the evaluation of natural suitability of human settlements based on topography, climate, hydrology and land cover conditions is the most widely used. And there is little study on human settlements on a large scale, and even more rarely in the Belt and Road regions. ”.

  • Line 73, when it comes to HSS, what is the main structure of HSS and how is it practically applied at present? This information is fundamental but missing.

Thank you very much for the careful review of the reviewer, who have supplemented the main structure of HSS and its current practical application according to your professional opinions. Manuscript revised to:

The study development of human settlements has gradually transitioned from the perspective of architecture and urban planning to the perspective of geography, and combined with 3S technology to establish a grid-scale evaluation index for the natural suitability of human settlements. Among them, the evaluation of natural suitability of human settlements based on topography, climate, hydrology and land cover conditions is the most widely used. And there is little study on human settlements on a large scale, and even more rarely in the Belt and Road regions. ”。

Evaluating the natural suitability of human settlements can help to not only scientifically determine human settlement conditions, which are affected by natural factors, but can also characterize regional population distribution patterns, which are affected by natural limitations or settlement suitability. Simultaneously, the results provide decision-making support for predicting population sizes, determining functional zones of population development, and ensuring sustainable development. ”.

In the Materials and Methods section,

  • Line 108, considering the widespread of your research area, it would be better to point out what is the quantity of the stations, do these stations spread evenly or satisfied for the interpolation resolution of this research?

Many thanks for the expert opinion. The description of the interpolation site has been supplemented at the corresponding position in the paper to fully meet the needs of large-scale research. Manuscript revised to:

The multi-year monthly average meteorological data in the study area are calculated by Kriging interpolation based on the GIS platform, using 2363 weather stations, with a spatial resolution of 1 km.

  • Lines 132 to 134, why you choose these four indices? you can add some explanations in the introduction part.

Many thanks to the reviewer for your careful review. According to expert opinions, the reasons for selecting indicators have been supplemented at the corresponding positions in the manuscript. Manuscript revised to:

There are various factors affecting the suitability of human settlements, but the most fundamental and determining factors of other natural environment, the main factors that play a leading role in the natural environment of human settlements mainly include the degree of terrain fluctuation, hydrothermal and climatic conditions, and hydrological conditions, as well as the characteristics of land use and vegetation coverage that comprehensively reflect the natural conditions of the region.

In the discussion section:

  • Relevant research has been cited but not discussed. Please add some key points of these research and some analysis to address the importance and meaning of your research results.

I am very grateful to the reviewer for your professional opinions and suggestions. According to the revision opinions, the importance and significance of the research results have been supplemented in the manuscript. Manuscript revised to:

In a long period of time, compared with the two rigid limiting factors of terrain (high altitude) and climate (cold), hydrology and land cover are important breakthroughs in improving the suitability of human settlements. According to the results of the suitability and restricted zoning of human settlements along the BRI region, water conservancy projects and technical measures (such as drip irrigation) should be used to improve seasonal water shortages and improve land cover conditions, so as to improve the suitability of regional human settlements. At the same time, make the best use of the situation and guide the people in an orderly manner according to the time and place to gradually flow and migrate from areas with unsuitable human settlements to areas with critically suitable and suitable human settlements. Based on the evaluation results of human settlements, it is possible to coordinate urban and rural development according to local conditions, reasonably control the scale of cities and towns, reasonably increase residential settlements, guide the orderly development of the population, and promote the adaptation of population distribution to the suitability of the human settlements environment.

  • the recommendations, lines 472-474, seem cannot be well supported by the results in this research. Please supply more explanations.

I fully agree with the expert's professional opinion, so delete the corresponding conclusion and add explanation, and supply more explanations. Manuscript revised to:

In a long period of time, compared with the two rigid limiting factors of terrain (high altitude) and climate (cold), hydrology and land cover are important breakthroughs in improving the suitability of human settlements. According to the results of the suitability and restricted zoning of human settlements along the BRI region, water conservancy projects and technical measures (such as drip irrigation) should be used to improve seasonal water shortages and improve land cover conditions, so as to improve the suitability of regional human settlements. At the same time, make the best use of the situation and guide the people in an orderly manner according to the time and place to gradually flow and migrate from areas with unsuitable human settlements to areas with critically suitable and suitable human settlements. Based on the evaluation results of human settlements, it is possible to coordinate urban and rural development according to local conditions, reasonably control the scale of cities and towns, reasonably increase residential settlements, guide the orderly development of the population, and promote the adaptation of population distribution to the suitability of the human settlements environment.

Conclusion:

  • Please revise and refine your conclusion section to better answer the scientific questions you addressed at the beginning of the manuscript.

I am very grateful to the reviewer for your professional opinions. The conclusions have been reorganized to correspond to the three scientific issues mentioned above. The specific modification is:

(1)Find out the natural background conditions of the Belt and Road regions, such as topography, climate, hydrology and land cover. Manuscript revised to:

—The physical and geographical conditions varied relatively significantly between the BRI regions, which are mainly affected by altitude and climate. For example, the topography, climate and ground cover conditions of alpine regions such as the Tibetan Plateau restrict settlement, while the vast plains have a pleasant climate and rich land and water resources. In particular, the RDLS ranged from 0 to 19.9, with an average value of 1.05, with the value gradually decreased from the Himalayas, Tianshan Mountains to the surrounding regions. As for the THI, except for the Tibetan Plateau, the THI gradually decreased from the low-latitude regions in the south to the high-latitude regions in the north. The average LSWAI was approximately 0.33 and that values were clustered in space for the BRI regions. The multi-year average LCI was approximately 0.21. Low LSWAI and LCI regions were primarily concentrated in two types of zones, namely, alpine zones as well as hot and dry zones.

(2)According to the degree of influence of physical and geographical factors in the BRI regions for the human settlement environment, and the correlation between the factors, this study built a human settlement suitability index model. Manuscript revised to:

— This study developed a 1 km grid-based model of comprehensive evaluation for HSS (based on the RDLS, THI, LSWAI, and LCI). The natural suitability and limitations of potential human settlements in various countries and regions along the BRI were briefly evaluated. Based on the single-factor correlation analyses, the HSS model consists of a square formed by two 2D quadrants interlaced with the dominant factor as the coordinate origin, with a side length of 1. The NRDLS and NTHI, which were two extremely weakly correlated factors, were placed at diagonal corners of the square as leading factors. And the LSWAI and LCI were used as the endpoint of the coordinates as auxiliary factors. Thus, the HSS model consisted of two evaluation results, namely, the terrain and climatic suitability corrected by the hydrological and terrain indices.

(3)Based on the GIS platform, form a raster atlas for comprehensive evaluation of human settlements in the BRI regions, and conduct grading and zonal evaluation of the human settlements in the study area, providing a research basis for the rational distribution of the population along the Belt and Road. Manuscript revised to:

—The HSS evaluation results were obtained based on the HSS model for the BRI area and revealed that the HSS for the BRI area ranged from 0.07 to 1, with an average value of 0.66. Spatially, the HSS gradually increased from the northeastern and southwestern parts of the study area to the central study area. Based on the terrain and climate suitability evaluation results, as well as the spatial distribution characteristics of the LSWAI and LCI, the HSS in the BRI was classified into six levels, namely, unsuitable, critically unsuitable, critically suitable, generally suitable, moderately suitable, and highly suitable. In particular, the human settlements suitability is dominated by critically suitable and suitable conditions accounted for 63% of the BRI. Based on the above comprehensive zoning of natural suitability of human settlements, it provides a basis for the rational distribution of population and spatial planning in the BRI. The results of the study can be used to guide the migration of populations with unsuitable regions to critically suitable regions, as well as continuously improve the hydrology and land cover conditions in critically suitable regions, and maintain the natural ecological environment in suitable zones for human settlements, so as to realize the healthy life of human beings and the sustainable development of society.

Thank you for your efforts to improve our manuscript. We hope that our responses and the resulting changes will be acceptable, but we will be happy to work with you to resolve any remaining issues.

Reviewer 4 Report

The article covers very important aspects. It is developed interestingly and reliably.
The only remark, due to the very extensive area of analysis and diverse methods, the conclusions should include a reference or comparison to the results of other researchers. Due to the topic, it is also advisable to supplement the letter with publications of the GIS scope. 

Author Response

Reviewer 4

The article covers very important aspects. It is developed interestingly and reliably.
The only remark, due to the very extensive area of analysis and diverse methods, the conclusions should include a reference or comparison to the results of other researchers. Due to the topic, it is also advisable to supplement the letter with publications of the GIS scope. 

I am very grateful to the reviewer for your affirmation of the research content and research direction. The research conclusions are further sorted out based on expert opinions, and GIS-wide publications are supplemented to support the discussion.

Please see manuscript for details:

The physical and geographical conditions varied relatively significantly between the BRI regions, which are mainly affected by altitude and climate. For example, the topography, climate and ground cover conditions of alpine regions such as the Tibetan Plateau restrict settlement, while the vast plains have a pleasant climate and rich land and water resources. In particular, the RDLS ranged from 0 to 19.9, with an average value of 1.05, with the value gradually decreased from the Himalayas, Tianshan Mountains to the surrounding regions. As for the THI, except for the Tibetan Plateau, the THI gradually decreased from the low-latitude regions in the south to the high-latitude regions in the north. The average LSWAI was approximately 0.33 and that values were clustered in space for the BRI regions. The multi-year average LCI was approximately 0.21. Low LSWAI and LCI regions were primarily concentrated in two types of zones, namely, alpine zones as well as hot and dry zones.

This study developed a 1 km grid-based model of comprehensive evaluation for HSS (based on the RDLS, THI, LSWAI, and LCI). The natural suitability and limitations of potential human settlements in various countries and regions along the BRI were briefly evaluated. Based on the single-factor correlation analyses, the HSS model consists of a square formed by two 2D quadrants interlaced with the dominant factor as the coordinate origin, with a side length of 1. The NRDLS and NTHI, which were two extremely weakly correlated factors, were placed at diagonal corners of the square as leading factors. And the LSWAI and LCI were used as the endpoint of the coordinates as auxiliary factors. Thus, the HSS model consisted of two evaluation results, namely, the terrain and climatic suitability corrected by the hydrological and terrain indices.

The HSS evaluation results were obtained based on the HSS model for the BRI area and revealed that the HSS for the BRI area ranged from 0.07 to 1, with an average value of 0.66. Spatially, the HSS gradually increased from the northeastern and southwestern parts of the study area to the central study area. Based on the terrain and climate suitability evaluation results, as well as the spatial distribution characteristics of the LSWAI and LCI, the HSS in the BRI was classified into six levels, namely, unsuitable, critically unsuitable, critically suitable, generally suitable, moderately suitable, and highly suitable. In particular, the human settlements suitability is dominated by critically suitable and suitable conditions accounted for 63% of the BRI. Based on the above comprehensive zoning of natural suitability of human settlements, it provides a basis for the rational distribution of population and spatial planning in the BRI. The results of the study can be used to guide the migration of populations with unsuitable regions to critically suitable regions, as well as continuously improve the hydrology and land cover conditions in critically suitable regions, and maintain the natural ecological environment in suitable zones for human settlements, so as to realize the healthy life of human beings and the sustainable development of society.

Thanks again to the reviewer for your affirmation and review.
